# Association between screen time and obesity in US adolescents: A cross-sectional analysis using National Survey of Children's Health 2016–2017

**Chighaf Bakour**[1,2]*, **Fahad Mansuri**[1], **Courtney Johns-Rejano**[3], **Michelle Crozier**[1,4], **Ronee Wilson**[1,2], **William Sappenfield**[1,2]

**1** College of Public Health, University of South Florida, Tampa, Florida, United States of America, **2** University of South Florida, The Chiles Center, Tampa, Florida, United States of America, **3** Morsani College of Medicine, University of South Florida, Tampa, Florida, United States of America, **4** School of Global Health Management and Informatics, University of Central Florida, Orlando, Florida, United States of America

* cbakour@usf.edu

**Data Availability Statement:** Data used in this study came from the National Survey of Children's Health. Data can be requested from the Data

## Abstract

### Objective

This study examines the association between time spent watching TV, playing video games, using a computer or handheld device (screen time), and BMI among U.S. adolescents, and potential effect modification of these associations by sex, sleep duration, and physical activity.

### Methods

A secondary analysis of 10–17-year-old participants in the 2016–2017 National Survey of Children's Health was conducted. Multivariable logistic regression was used to examine the association between parent-reported screen time and BMI categories and effect modification by sex, sleep duration and physical activity.

### Results

The analysis included 29,480 adolescents (49.4% female). Those with ≥1 hour (vs <1 hour) of TV/video games per day were more likely to be overweight/obese (adjusted Odds Ratio (aOR) 1–3 hours = 1.4; 95% Confidence Interval (CI)1.19, 1.65; aOR ≥4 hours = 2.19; 95% CI 1.73, 2.77). This association was stronger in adolescents who did not meet the guidelines for physical activity (aOR ≥ 4 hours = 3.04; 95% CI: 2.1, 4.4) compared with those who did (aOR ≥ 4 hours = 1.64; 95% CI: 0.72, 3.72). Using computers/handheld devices was associated with a smaller increase in odds of overweight/obesity (aOR ≥4 hours = 1.53; 95% CI:1.19, 1.97).

### Conclusion

Watching TV or playing video games for ≥1 hour per day is associated with obesity in adolescents who did not meet the guidelines for physical activity. Using computers or

Resource Center for Child and Adolescent Health: https://www.childhealthdata.org/help/dataset.

**Funding:** The author(s) received no specific funding for this work.

**Competing interests:** The authors have declared that no competing interests exist.

**Abbreviations:** TV, television; OR, odds ratio; CI, confidence interval; BMI, body mass index; U.S., United States; NSCH, National Survey of Children's Health (NSCH).

handheld devices seems to have a weaker association with BMI compared with TV/video games.

## Introduction

Childhood and adolescent obesity is a global public health crisis with one in five youth estimated to be overweight or obese [1]. Childhood obesity is a serious problem in the US, putting children and adolescents at risk for poor health [2]. In the U.S. between 2017 and 2020, the prevalence of obesity amongst children aged 6- to 11- years and adolescents aged 12- to 19-years was 20.7%, and 22.2%, respectively [3]. Obesity in childhood and adolescence increases the risk of many outcomes including obesity, hypertension, diabetes, asthma, sleep apnea, and dyslipidemia in adulthood [2, 4, 5]. Thus, it is paramount to identify emerging risk factors of obesity in adolescents that may be targeted with early interventions to prevent negative health outcomes in adulthood [4, 5].

Lack of physical activity, poor sleeping and eating habits, consumption of sugary beverages, and a sedentary lifestyle are important risk factors for overweight/obesity in adolescents [5–9]. Time spent in front of screens such as television, computer, or hand held devices has also been implicated as a risk factor for obesity, as it is commonly accompanied by poor eating habits, sedentary behavior, and lack of physical activity [10]. A recent study found that more than 65% of adolescents in the US exceed the recommended amount of screen time of no more than two hours of recreational screen time per day [11]. Screen time has steadily increased among adolescents, and the proportion of teens (ages 14–17) reporting being online almost constantly has doubled from 24% in 2015 to 45% in 2018 [12]. A meta-analysis of 52 studies (n = 44,707) found that TV viewing was positively associated with body fatness in children and adolescents [13], and another meta-analysis (16 studies, n = 17,920) published in 2019 found that screen time of two or more hours per day was associated with 67% (Odds Ratio (OR) = 1.67 95% Confidence Interval (CI): 1.48–1.88) higher odds of overweight/obesity in children and adolescents [10]. Similar positive associations between excessive screen time and overweight/obesity in adolescents have been reported by numerous cross-sectional [14–20] and longitudinal [21–23] studies in different ethnicities and populations. Thus, there is ample evidence showing that excessive screen time is positively associated with obesity/overweight in adolescents. However, some questions remain to be answered, especially regarding the difference in the association between males and females, which was examined in few studies with inconsistent findings [29, 31, 34, 35]. Additionally, the potential interaction between screen time and other risk factors for obesity, namely physical activity and sleep duration, has not been adequately explored in adolescents, although a few studies indicated that adequate physical activity may attenuate the effects of increased screen time [32, 37, 38].

Screen-based activities include watching television, playing video games, using a computer, or using a handheld device like a mobile phone, a tablet, or a handheld video game. This brings up the question of whether time spent in front of computers compared with time spent watching television or playing video games has a similar effect on weight. Few studies have examined differences by type of screen-related activity with inconsistent findings, with some indicating that time spent on TV but not on computer is associated with obesity [31–33], while others showing an association between computer time and BMI [34]. The difference in results may be due to the difference in population characteristics, as the first three studies that found no association between computer use and BMI conducted in Germany, Brazil, and Finland, while the study that found an association was conducted in the US.

This study examines the association between two separate types of screen time (using a computer or a handheld device; and watching TV or playing video games) and body mass index (BMI) among US adolescence, as well as difference in these associations by sex. Additionally, we examine the effect medication of meeting daily recommendations for sleep duration and physical activity on the association between screen time and BMI.

## Methods

Data for this analysis were derived from the 2016–2017 National Survey of Children's Health (NSCH), which is a nationally representative survey of U.S. parents and caregivers regarding the health and wellbeing of their non-institutionalized children aged 0–17 years. The survey is sponsored by the Health Resources and Services Administration's Maternal and Child Health Bureau and conducted annually by the U.S. Census Bureau using mailed invitations sent to randomly selected households. The survey can be completed either by mail or online. A full description of the survey methods can be found elsewhere [24]. This analysis includes adolescents aged 10–17 years whose parents responded to the 2016 or 2017 surveys and who have complete data on BMI and screen time (N = 35,580). Adolescents with moderate to severe autism; brain injury; heart disease; down syndrome; genetic disorders; cerebral palsy; blindness; or significant functional limitation were excluded from the analysis, leading to a final sample size of 29,480. This study received an exemption determination by the Institutional Review Board at the University of South Florida.

### Measures

**Outcome variable.** *Obesity* was measured by BMI which was calculated based on the child's height and weight reported by parents. BMI was categorized using age and sex-specific percentiles according to Centers for Disease Control and Prevention (CDC) growth charts as: underweight (less than 5th percentile); healthy weight (5th to 84th percentile); overweight or obese (85th percentile or above). Healthy weight was used as the reference value.

**Exposure variable.** *Screen time* was defined by two different measures constructed using the following questions: 1) On an average weekday, about how much time does this child usually spend with computers, cell phones, handheld video games, and other electronic devices, doing things other than schoolwork? and, 2) On an average weekday, about how much time does this child usually spend in front of a TV watching TV programs, videos, or playing video games? Response options included none, less than one hour, 1 hour, 2–3 hours, 4 hours or more. For the purpose of this analysis, both screen time variables were categorized as: <1 hour per day (reference category), 1–3 hours per day, and ≥4 hours per day.

**Effect modifiers.** *Sleep duration* was measured by the parent's response to the question "During the past week, how many hours of sleep did this child get on an average weeknight?" The sleep duration variable was constructed according to the American Academy of Sleep Medicine's consensus statement on recommended sleep required for optimal health. According to the statement the recommended sleep duration for children aged 6–12 and 13–18 years is 9–12 hours and 8–10 hours respectively [25]. Based on this recommendation, the sleep duration was divided into two categories: child sleeps recommended age-appropriate hours and child sleeps less than recommended age-appropriate hours.

*Physical activity* was assessed based on the parent's response to the question "during the past week, on how many days did this child exercise, play a sport, or participate in physical activity for at least 60 minutes?" The U.S. Department of Health and Human Services' physical activity guidelines for adolescents [26] recommend at least 60 minutes of physical activity per

day, and the variable was categorized as: does not meet the recommended physical activity guidelines or meets the recommended physical activity guidelines.

*Covariates* were measured based on responses by the parent or guardian of the child, and included child's age (years), sex (male or female), race/ethnicity (Hispanic, White non-Hispanic, Black non-Hispanic, multi-racial/other non-Hispanic), child health status (excellent, good, fair or poor); depression status (does not have depression, ever told but not current, current depression); anxiety status (does not have anxiety, ever told but not current, current anxiety); and ADHD severity (does not currently have ADHD, mild ADHD, moderate or severe ADHD), in addition to family's income compared with the federal poverty level FPL (0–99% FPL, 100%-199% FPL, 200%-399% FPL, 400% FPL or above).

## Statistical analysis

Frequency and percentage were used to describe the distribution of categorical variables and median and intra-quartile range was used to describe the distribution of age for the entire sample and by BMI. The distributions of categorical and continuous variables were calculated using SAS 9.4 (Cary, NC) procedures, Survey Freq and Survey means, respectively. These procedures incorporate complex survey designs, including designs with stratification, clustering, and unequal weighting Multinomial logistic regression analysis was used to examine the association of each of the screen time variables (TV or video games, computer or handheld device) with BMI categories. The reference group for BMI was normal weight and the reference group for screen time variables was screentime of <1 hour per day. For both screen time variables, we started with an unadjusted model, followed by a multivariable model adjusted for age, sex, race, health status, depression status, anxiety status, ADHD severity and family income. The models were then stratified by sex to explore potential effect heterogeneity by sex. To examine potential effect modification by sleep duration and physical activity, we added two interaction terms separately (screen time*sleep) and (screen time*physical activity) to each of the fully adjusted models, to examine the combined effect of screen time and each of the two variables, compared with the reference value (screen time <1 hour, meets the guidelines of sleep duration/physical activity).These analyses were repeated for the fully adjusted model stratified by sex. Odds ratios (OR) and 95% confidence intervals (CI) were calculated for all associations. The multinomial logistic regression analysis was performed using SAS 9.4 (Carey, SC) Survey logistic procedure.

## Results

The study included 29,480 adolescents (49.4% female), with a median age of 13 years. About 30.7% of the participants were overweight/obese, while 6.3% were underweight. Adolescents males had higher prevalence of overweight/obese (32.3% vs 29.0% male) and underweight (7.3% vs 5.0%52.2% male) as compared to adolescent females. Prevalence of overweight/obesity was higher in Hispanic (39.3%) and Black (39.4%) adolescents as compared to white adolescents (29.0). Prevalence of overweight/obesity was higher in adolescents living in poverty (FPL: 0–99%, 38.4%), with fair or poor general health (61.7%), depression (38.4), anxiety (34.9%). The prevalence of overweight/obesity was higher in adolescents who did not get recommended amount of sleep (31.8% vs 25.7%) and physical activity (31.8% vs 25.7%). (Table 1).

Adolescents spending ≥4 hours with a computer or handheld device had the highest prevalence of overweight/obesity (36.6%) compared with 29.5% in adolescents spending 1–3 hours and 27.5% in adolescents spending <1 hour with a computer or handheld device. A similar pattern was observed in time spent watching TV or playing video games. (Table 1).

**Table 1. Characteristics of adolescent participants in the National Survey of Children's Health.**

| | Underweight | Healthy weight | Overweight or obese |
|---|---|---|---|
| | N (%) | N (%) | N (%) |
| | 1825 (6.3) | 19873 (63.1) | 7782 (30.7) |
| Age, Median (IQR)* | 12 (10–14) | 13 (11–15) | 13 (11–15) |
| **Sex** | | | |
| • Male | 1013 (7.3) | 9227 (60.4) | 4232 (32.3) |
| • Female | 812 (5.3) | 10646 (65.752.6) | 3550 (29.0) |
| **Race** | | | |
| • Hispanic | 202 (5.7) | 1876 (55.1) | 1092 (39.3) |
| • White, non-Hispanic | 1258 (6.5) | 14455 (68.6) | 5096 (25.0) |
| • Black, non-Hispanic | 94 (4.8) | 1008 (55.8) | 668 (39.4) |
| • Multi-racial/Other, non-Hispanic | 271 (8.9) | 2534 (64.5) | 926 (26.6) |
| **Poverty Level** | | | |
| • 0–99% FPL | 173 (6.9) | 1503 (54.9) | 946 (38.2) |
| • 100%-199% FPL | 260 (5.6) | 2525 (58.4) | 1405 (36.0) |
| • 200%-399% FPL | 502 (5.5) | 5934 (63.2) | 2562 (31.3) |
| • 400% FPL or above | 890 (7.0) | 9911 (7.07) | 2869 (22.3) |
| **Child Health Status** | | | |
| • Excellent or very good | 1712 (6.4) | 18961 (65.4) | 6785 (28.2) |
| • Good | 96 (5.0) | 791 (42.3) | 884 (52.7) |
| • Fair or poor | 11 (5.0) | 75 (33.3) | 83 (61.7) |
| **Anxiety** | | | |
| • Never | 1629 (6.2) | 17827 (63.4) | 6790 (30.4) |
| • Ever, but not current | 32 (16.8) | 319 (54.9) | 132 (28.3) |
| • Current | 150 (4.7) | 1610 (60.4) | 800 (34.9) |
| **Depression** | | | |
| • Never | 1750 (6.4) | 18805 (63.4) | 7114 (30.2) |
| • Ever, but not current | 16 (3.1) | 257 (48.7) | 121 (48.2) |
| • Current | 55 (4.4) | 727 (57.2) | 501 (38.4) |
| **ADHD** | | | |
| • No current ADHD | 1592 (6.3) | 17892 (63.0) | 6912 (30.7) |
| • Current mild | 118 (7.8) | 977 (62.7) | 402 (29.5) |
| • Current moderate or severe | 87 (4.9) | 790 (62.5.) | 382 (32.6) |
| **Physical Activity** | | | |
| • Meets recommendations | 339 (5.5) | 3754 (68.8) | 1097 (25.7) |
| • Does not meet recommendations | 1452 (6.4) | 15776 (61.8) | 6546 (31.8) |
| **Sleep** | | | |
| • Meets recommendations | 1279 (6.8) | 13429 (64.9) | 4746 (28.3) |
| • Does not meet recommendations | 508 (5.2) | 6102 (60.1) | 2904 (34.7) |
| **Time Spent with Computers/ Handheld Device** | | | |
| • None or less than 1 hour per day | 290 (7.6) | 2470 (64.9) | 802 (27.5) |
| • 1–3 hours per day | 1223 (6.4) | 13755 (64.1) | 5141 (29.5) |
| • 4 hours or more per day | 296 (5.0) | 3487 (58.4) | 1773 (36.6) |
| **Time Spent Watching TV /Playing Video Games** | | | |
| • None or less than 1 hour per day | 394 (6.3) | 4505 (71.6) | 1226 (22.1) |
| • 1–3 hours per day | 1255 (6.5) | 13622 (62.4) | 5440 (31.1) |
| • 4 hours or more per day | 162 (5.2) | 1582 (51.0) | 1043 (43.8) |

Table 2 shows the adjusted ORs of the association between time spent watching TV or playing video games and having overweight/obesity. It shows that spending 1–3 hours (vs. <1 hour) watching TV or playing video games was associated with a 40% increase in the odds of having overweight/obesity (Adjusted OR = 1.4, 95% CI 1.19, 1.65). Similarly, spending ≥4 hours (vs <1 hour) watching TV or playing video games was associated with 2.19 times likelihood of having overweight/obesity (adjusted OR = 2.19, 95% CI 1.73, 2.77). The association between time spent watching TV or playing video games and having overweight/obesity stratified by sex also showed similar results (Table 2). Table 3 shows the association between time

**Table 2. Results of logistic regression analysis of the association between time spent watching TV or playing video games and BMI in adolescents.**

| Hours spent watching TV or playing video games per day | Overweight or obese vs. Normal BMI | | | | | |
|---|---|---|---|---|---|---|
| | Total | | Males | | Females | |
| | Unadjusted OR (95% CI) | Adjusted OR (95% CI)* | Unadjusted OR (95% CI) | Adjusted OR (95% CI)* | Unadjusted OR (95% CI) | Adjusted OR (95% CI)* |
| None or less than 1 hour per day | 1.00 | 1.00 | 1.00 | 1.00 | 1.00 | 1.00 |
| 1–3 hours per day | 1.61 (1.37, 1.9) | 1.4 (1.19, 1.65) | 1.42 (1.09, 1.86) | 1.33 (1.03, 1.71) | 1.70 (1.39, 2.09) | 1.45 (1.18, 1.78) |
| 4 hours or more per day | 2.78 (2.19, 3.52) | 2.19 (1.73, 2.77) | 2.20 (1.57, 3.10) | 1.92 (1.39, 2.63) | 3.72 (2.56, 5.41) | 2.56 (1.82, 3.61) |

*Adjusted for age, sex, race, health status, family income, depression, anxiety, ADHD, and amount of sleep and physical activity.

spent using a computer or handheld device and having overweight/obesity for all participants and separately for males and females. Using a computer or handheld device was associated with having overweight/obesity in adolescents who spent ≥4 hours (vs <1 hour) on these devices (OR = 1.53, 95% CI 1.19, 1.97). Spending ≥4 hours (vs <1 hour) using a computer or handheld device was associated with having overweight/obesity in adolescent females (OR = 1.71, 95% CI 1.2, 2.44), but not in adolescent males (OR = 1.4, 95% CI 0.99, 1.99).

Table 4 shows the association between time spent watching TV or playing video games and having overweight/obesity by physical activity, with adolescents spending <1 hour watching TV or playing video games and meeting the guidelines for recommended physical activity as the reference. Adolescents spending 1–3 hours watching TV or playing video games and not meeting the guidelines for recommended physical activity were 1.88 (95% CI: 1.34, 2.62) times likely to have overweight/obesity. Likewise, adolescents spending ≥4 hours watching TV or playing video games and not meeting the guidelines for recommended physical activity were 3.04 (95% CI:2.10, 4.40) times more likely to have overweight/obesity. Adolescent females not meeting the guidelines for required physical activity and spending <1 hour, 1–3 hours or ≥4 hours watching TV or playing video games were 1.71 (95% CI: 1.03, 2.84), 2.40 (95% CI: 1.48, 3.9) and 4.28 (95% CI: 2.45, 7.49) times likely to have overweight/obesity, respectively (Table 4). Similarly, adolescent males not meeting the guidelines for required physical activity and spending 1–3 hours or ≥4 hours watching TV or playing video games were 1.57 (95% CI: 1.01, 2.45), and 2.35 (95% CI:1.45, 3.81) times likely to have overweight/obesity, respectively (Table 4).

Table 5 shows the association between hours spent with a computer or handheld device and having overweight/obesity. Adolescents spending ≥4 hours with a computer or handheld device and not meeting the guidelines for recommended physical activity were 1.71 (95% CI: 1.08, 2.70) times likely to have overweight/obesity compared with adolescents spending <1

**Table 3. Results of logistic regression analysis of the association between time spent with computers, cell phones, handheld video games and other electronic devices, and BMI in adolescents.**

| Hours spent with a computer or handheld device per day | Overweight or obese vs. Normal BMI | | | | | |
|---|---|---|---|---|---|---|
| | Total | | Males | | Females | |
| | Unadjusted OR (95% CI) | Adjusted OR (95% CI)* | Unadjusted OR (95% CI) | Adjusted OR (95% CI)* | Unadjusted OR (95% CI) | Adjusted OR (95% CI)* |
| None or less than 1 hour per day | 1.0 | 1.0 | 1.0 | 1.0 | 1.0 | 1.0 |
| 1–3 hours per day | 1.09 (0.88, 1.34) | 1.17 (0.94, 1.47) | 1.11 (0.82, 1.49) | 1.18 (0.86, 1.61) | 1.06 (0.79, 1.44) | 1.2 (0.88, 1.64) |
| 4 hours or more per day | 1.48 (1.17, 1.87) | 1.53 (1.19, 1.97) | 1.32 (0.95, 1.85) | 1.4 (0.99, 1.99) | 1.69 (1.21, 2.37) | 1.71 (1.2, 2.44) |

[a] Adjusted for age, sex, race, health status, family income, depression, anxiety, ADHD, and amount of sleep and physical activity.

**Table 4. Results of logistic regression analysis of the effect modification by physical activity on the association between time spent watching TV or playing video games and overweight.**

| | | Overweight or obese vs. Normal BMI | | |
|---|---|---|---|---|
| | | Total | Males | Females |
| Physical Activity | Hours spent watching TV or playing video games per day | Adjusted OR (95% CI)* | Adjusted OR (95% CI)* | Adjusted OR (95% CI)* |
| Meets guidelines | None or less than 1 hour per day | 1.00 | 1.00 | 1.00 |
| Meets guidelines | 1–3 hours per day | 1.36 (0.93, 1.99) | 1.14 (0.69, 1.88) | 1.74 (0.99, 3.03) |
| Meets guidelines | 4 hours or more per day | 1.64 (0.72, 3.72) | 1.34 (0.54, 3.33) | **3.02 (1.04, 8.75)** |
| Does not meet guidelines | None or less than 1 hour per day | 1.33 (0.93, 1.91) | 1.12 (0.67, 1.85) | **1.71 (1.03, 2.84)** |
| Does not meet guidelines | 1–3 hours per day | **1.88 (1.34, 2.62)** | **1.57 (1.01, 2.45)** | **2.40 (1.48, 3.9)** |
| Does not meet guidelines | 4 hours or more per day | **3.04 (2.10, 4.40)** | **2.35 (1.45, 3.81)** | **4.28 (2.45, 7.49)** |

[a] Adjusted for age, sex, race, health status, family income, depression, anxiety, ADHD, and amount of sleep and physical activity.

hour with a computer or handheld device and meeting the guidelines for recommended physical activity. Adolescent females spending ≥4 hours with a computer or handheld device and not meeting the guidelines for recommended physical activity were twice likely to have overweight/obesity (OR = 2.15, 95% CI 1.08, 4.28) as compared to adolescent females spending <1 hour with a computer or handheld device and meeting the guidelines for recommended physical activity.

Time spent watching TV or playing video games was significantly associated with overweight regardless of whether the child sleeps for the recommended time. Similar patterns were observed when examining the interaction between sleep and the use of computer/handheld device variable (S1 Table).

## Discussion

Using recent data from the National Study of Children's Health, we sought to quantify the relationship between time spent on different types of screens and overweight/obesity in adolescence, and whether this relationship differs by sex, physical activity, or sleep duration. Overall, our findings support the hypothesis that time spent on screens is associated with overweight/obesity, however, the association is more pronounced when the time is spent watching television or playing video games than it is when using computers or handheld devices. Physical activity, and to a lesser extent, sex, appear to modify this association, while sleep duration does not. Our findings add to previous studies that examined the association between screen time

**Table 5. Results of logistic regression analysis of the effect modification by physical activity on the association between time spent using a computer or handheld device and overweight.**

| | | Overweight or obese vs. normal BMI | | |
|---|---|---|---|---|
| | | Total | Males | Females |
| Physical Activity | Hours spent with a computer or handheld device per day | Adjusted OR (95% CI)* | Adjusted OR (95% CI)* | Adjusted OR (95% CI)* |
| Meets guidelines | None or less than 1 hour per day | 1.00 | 1.00 | 1.00 |
| Meets guidelines | 1–3 hours per day | 0.86 (0.54, 1.38) | 0.79 (0.43, 1.43) | 1.00 (0.49, 2.08) |
| Meets guidelines | 4 hours or more per day | 0.76 (0.41, 1.43) | 0.78 (0.36, 1.71) | 0.72 (0.29, 1.77) |
| Does not meet guidelines | None or less than 1 hour per day | 0.94 (0.59, 1.53) | 0.82 (0.44, 1.53) | 1.12 (0.54, 2.31) |
| Does not meet guidelines | 1–3 hours per day | 1.24 (0.80, 1.92) | 1.16 (0.65, 2.04) | 1.41 (0.72, 2.75) |
| Does not meet guidelines | 4 hours or more per day | **1.71 (1.08, 2.70)** | 1.42 (0.79, 2.58) | **2.15 (1.07, 4.28)** |

[a] Adjusted for age, sex, race, health status, family income, depression, anxiety, ADHD, and amount of sleep and physical activity.

and obesity in adolescence and provides additional evidence regarding different types of screen activity and the potential role of adequate physical activity in ameliorating some of the negative effects of increased screen time.

Although we are unable to claim a causal relationship between screen time and overweight/ obesity, our findings, combined with prior research, highlight the importance of adequate physical activity for adolescents who spend considerable amounts of time engaged with various types of screens. The highest odds of overweight or obese BMI were in adolescents, particularly females, who spent four or more hours on screens and who do not meet the recommended guidelines for physical activity. Adolescents who met the recommended time for physical activity had no increase in the odds of having overweight or obesity even with four or more hours of screen time, except females with four or more hours of TV/video games time, in whom the odds were attenuated with physical activity, but remain elevated. Encouraging daily physical activity, accompanied by a healthy diet, can be important and effective strategy to combat the adverse effects of prolonged hours spent sitting in front of a computer or using a tablet. Moreover, with the shift in screen time habits in adolescents in recent years, and more time being spent using computers and handheld devices compared with television viewing [28], our results add to the much-needed knowledge regarding the impact of these new types of screen activities on adolescent BMI. Furthermore, with the difficulty in limiting the time that adolescents spend in front of screens, whether for education, socializing, or leisure, it is now more important than ever to identify factors that reduce the negative effects of screen time, in particular by encouraging adequate physical activity.

Our findings are consistent with prior research showing an association between screen time or TV viewing and obesity among adolescents [10, 27, 28], and with some studies that examined computer time separately [29–31], most of which found little to no association between time spent using a computer and overweight in adolescence. Most of these studies were conducted in children under the age of 12, with only few including adolescents. By contrast, a study by Utter et al. (2003) [34] of high school students found that high computer use (four or more hours per day) was associated with higher mean BMI in boys. However, this study did not adjust for physical activity or health status, and was conducted during the 1998–1999 school year, when access to computers and the internet was still limited and participants spent an average of 1–1.25 hours per day on computers. Our present study using recent data is more representative of current screen time habits in adolescents.

Differences related to screen activity type may be explained by other activities done at the same time, such as eating while watching television, which is less likely when using the computer [32]. Additionally, TV viewing is a passive activity that often takes an hour or more at a time, while most computer use requires active interaction, and may be done for short but recurring periods. Moreover, TV viewing exposes adolescents to unhealthy food advertising, which may affect their dietary habits [33]. Along with sedentary behavior, these factors may contribute to overweight. It is important to note however, that TV viewing in the present study was combined with video game playing in one variable, which prevents us from making conclusions regarding which activity may have had the stronger association with BMI. Likewise, computer use was combined with use of handheld devices, which unlike computers, may be used while sitting, standing, or walking.

Sex-related differences in the association between screen time and BMI observed in our study were somewhat similar to the results observed by Kautiainen et al. [31], who found a significant association between screen time and BMI in 14–18 year-old girls but not in boys. Slightly different pattern was observed in Utter et al's [34] study of middle and high school students, which found TV viewing to be significant in males and females, and computer time to be significant only in females. Likewise, Te Velde et al. [29] found a significant association

between screen time and BMI in 9–14 year-old males and females. The three studies mentioned above were conducted more than 10 years prior to our data collection, which may explain some of the observed differences due to the change in types of screen activities in adolescents over time. Furthermore, differences in results may be due to different age group, measurement of screen time, and statistical methods used in each study. By contrast, a recent study by Sampasa-Kanyinga et al. (2020) found the association between social media use and BMI to be significant in males in 7th-12th grade, but not in females of the same age [35]. Nevertheless, the pattern observed in our study is consistent with studies in adults, which found a stronger association between time spent watching TV and metabolic syndrome in women than in men [36, 37]. More research is needed to understand the impact of different types of screen time on BMI in males and females.

The observed effect modification by physical activity is similar to what Mutunga et al. [38] observed, but in their study, the attenuating effect of increased physical activity was only observed in adolescents with high socioeconomic status. Likewise, a recent study of Canadian children aged 7–12 found that sedentary time was significantly associated with BMI only in children who do not meet the physical activity guidelines [39]. Additionally, the systematic review by Rey-Lopez and colleagues [30] concluded that unlike TV viewing, "playing video games and using computers do not represent such a high risk if these do not replace too much physical activity", indicating potential effect modification by physical activity. Although we are not able to determine whether screen time is replacing physical activity in our cross-sectional study, this may be a plausible explanation of the effect modification we observed in our study, considering that adolescents who find the time for daily physical activity despite spending ≥4 hours on screens have significantly lower odds of obesity compared with those who are less frequent active. Furthermore, the energy expenditure associated with daily physical activity compensates, at least partially, for the sedentary periods spent on screens.

Our study has many strengths, including the large representative sample of US adolescents, use of recent data examining new types of technology and current media use behavior, and the examination of different types of screen activities in males and females and effect modification by physical activity. Nonetheless, there are some limitations that must be noted. The cross-sectional design does not permit the determination of causality between time spent on screens, physical activity, and BMI; therefore, our findings need to be confirmed with longitudinal studies. Additionally, time spent on screens was reported by parents of the adolescents, which may lead to misclassification of the exposure. A study of Canadian children aged 9–11 found that parental and child reports or TV viewing time were generally concordant, however parents consistently reported less TV exposure than children [40]. Although this study was in a younger age group, it is logical to assume that parental report will generally underestimate screen time in adolescent, given the increasing independence of adolescents as they age. As long as this underestimation is independent of weight status, it will lead to an underestimation of the measure of association. Likewise, height and weight were reported by parents, which may lead to underestimation of BMI, especially in females [41], although a study by Goodman et al. [42] found parental report of height and weight is better than teen report, and correctly classifies 96% as to obesity status. One important limitation is the combination of TV watching and video game use in one variable. Television viewing is mostly a passive activity that does not lead to energy expenditure and is frequently accompanied by eating or snaking. By contrast, playing video games requires physical interaction with the controls, which may not allow eating or snacking, and some games include some physical activity (active video games). Therefore, we are unable to determine the impact of television viewing and video game playing separately. Likewise, computers and handheld devices may differ in the context of their use and their impact on weight. Furthermore, the question regarding time spent on computers

specified time doing activities other than schoolwork, which limits the generalizability of our findings. Finally, this study uses data collected in 2016–2017, which may not be reflective of current screen-based practices, especially considering the Covid-19 pandemic and the shift to online education.

## Conclusion

The results of our study indicate that time spent in front of screens, is significantly associated with obesity in US children and adolescents. Our findings add to the literature by showing that this association is stronger in females compared with males, and when watching television or playing video games compared with computers or handheld devices, but weaker in those with 60 minutes or more of physical activity every day. Given the increasing utilization of screen-based activities, it is important to encourage adequate physical activity in adolescents, particularly females, which may ameliorate some of the effects of extended screen time. Future research is needed to confirm the findings and to test interventions that encourage reduced screen time and increased physical activity among adolescents.

## Supporting information

**S1 Table. Results of logistic regression analysis of the effect modification by sleep duration on the association between time spent watching TV or playing video games and overweight.**
(DOCX)

## Author Contributions

**Conceptualization:** Chighaf Bakour.

**Data curation:** Ronee Wilson.

**Formal analysis:** Chighaf Bakour, Fahad Mansuri.

**Methodology:** Chighaf Bakour, Michelle Crozier, Ronee Wilson, William Sappenfield.

**Writing – original draft:** Chighaf Bakour, Fahad Mansuri, Courtney Johns-Rejano.

**Writing – review & editing:** Chighaf Bakour, Michelle Crozier, Ronee Wilson, William Sappenfield.

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
