## [Decision Letter · Decision Letter 0]

1 Aug 2022

PONE-D-22-15444Association between screen time and obesity in US adolescents: A cross-sectional analysis using National Survey of Children’s Health 2016-2017PLOS ONE

Dear Dr. Bakour,

Thank you for submitting your manuscript to PLOS ONE. After careful consideration, we feel that it has merit but does not fully meet PLOS ONE’s publication criteria as it currently stands. Therefore, we invite you to submit a revised version of the manuscript that addresses the points raised during the review process.

We look forward to receiving your revised manuscript.

Kind regards,

Yih-Kuen Jan, PhD

Academic Editor

PLOS ONE

Journal Requirements:

2. You indicated that ethical approval was not necessary for your study. We understand that the framework for ethical oversight requirements for studies of this type may differ depending on the setting and we would appreciate some further clarification regarding your research. Could you please provide further details on why your study is exempt from the need for approval and confirmation from your institutional review board or research ethics committee (e.g., in the form of a letter or email correspondence) that ethics review was not necessary for this study? Please include a copy of the correspondence as an ""Other"" file.

Reviewers' comments:

Reviewer's Responses to Questions

**Comments to the Author**

1. Is the manuscript technically sound, and do the data support the conclusions?

Reviewer #1: Yes

Reviewer #2: Yes

2. Has the statistical analysis been performed appropriately and rigorously? 

Reviewer #1: Yes

Reviewer #2: No

3. Have the authors made all data underlying the findings in their manuscript fully available?

Reviewer #1: Yes

Reviewer #2: No

4. Is the manuscript presented in an intelligible fashion and written in standard English?

Reviewer #1: Yes

Reviewer #2: Yes

5. Review Comments to the Author

Reviewer #1: This manuscript examines associations between screen-use and obesity, and examines ineractions with physical activity and sleep in a large sample of adolescents from the US. While this is a generally well written manuscript, there are a number of comments for your consideration below:

Abstract:

A little untidy with some mistakes/incomplete information. For example, in the introduction you state that you examine effect modification by sleep and PA, then in the methods you state sex and pa. furthermore, you examine the associations between screen time and BMI not the other way around. Consistency in ordering is necessary for the reader to understand what you have done. Nothing in the methods section about how behaviours etc were measured. Results are missing the n and seem quite brief. Consider stating pa not exercise as you measured whether or not adolescents met the pa guidelines, not exercise.

Introduction:

I found this section lacking in a clear and concise rational.

The data you present on obeisty is quite out of date.

The second paragraph is very disjointed with talk of screen time then sedentary behaviour, but no definition or distinction described.

No rational for sleep or pa as effect modifiers.

I don't think the rational around covid is appropriate here given that your data preceed covid.

Methods:

This section is relatively well described.

We aren't told how covariates were measured and given that you describe them in results, the reader should know how they were measured.

Statistical analysis section is very brief. i would expect more on why you chose this method to look at effect modification and more descripton. also the ordering seems strange. Typically you would see if your main variables differ by gender at the start and then all subsequent analyses are conducted by gender.

Results:

This section is too long and the tables could be better presented by having gender as columns not extra rows where variabls are repeated.

First paragraph has a mistake in th % in second sentence.

Discussion:

Lacking in discussion and implicatations.

Limitations need expanding to add in that this data is >6 years old and not reflective of current screen-based practices.

Using the phrase exercise is unclear.

Bottom paragraph on page 19 is unclear when you are talking of replacement of pa by screen-time. Your analysis doesnt allow such statements to be made. you could allude to isotemporal substitution, but not suggest that that is what is hapenning here.

No conclusion?

what do your findings mean, to whom and what shall we do now that we have them?

Reviewer #2: Bakour and colleagues analyzed the 2016-2017 National Survey of Children’s Health and assessed the association between time spent watching TV, playing video games, using a computer or handheld device, and BMI among U.S. adolescents and potential effect modification of these associations by sleep duration and physical activity. The study has merit but I have several questions/comments:

1. Abstract: Please elaborate what does aOR mean? Then you can use the abbreviation.

2. Abstract: Line 39: Please write ‘95% CI’ before using 95% confidence interval.

3. After using an odds ratio, please put a semicolon before mentioning 95% CI. For example in Line 39 please use 1.53; 95% CI: 1.19, 1.97.

4. Before using Ordinal logistic regression, was proportional odds assumption verified by the authors?

5. In Table 1, please use row percentages instead of column percentages. That will give some idea about the prevalence of each weight categories.

6. In table 2 and 3, please report the unadjusted odds for male and female.

7. In Page 14, what is the title of the table?

8. In Table 3, why significant association was found for females but not for males? Please discuss.

6. PLOS authors have the option to publish the peer review history of their article (what does this mean?). If published, this will include your full peer review and any attached files.

Reviewer #1: No

Reviewer #2: **Yes: **Rajat Das Gupta, University of South Carolina

---

## [Author Response · Author response to Decision Letter 0]

15 Sep 2022

Journal Requirements:

Response: Thank you for your comment. The manuscript meets PLOS ONE’s style requirements. Please let us know if additional changes are necessary.

2. You indicated that ethical approval was not necessary for your study. We understand that the framework for ethical oversight requirements for studies of this type may differ depending on the setting and we would appreciate some further clarification regarding your research. Could you please provide further details on why your study is exempt from the need for approval and confirmation from your institutional review board or research ethics committee (e.g., in the form of a letter or email correspondence) that ethics review was not necessary for this study? Please include a copy of the correspondence as an ""Other"" file.

Response: This study was part of a larger project that included secondary analyses of the National Survey of Children’s Health. The project title was “Influence of social determinants on child health: A secondary analysis of the National Survey of Children’s Health”, PI: Dr. Ronee Wilson, a co-author on this manuscript. The project was submitted to USF IRB in 2018 and received an exempt status (does not meet the definition of human subject research). IRB#: Pro00035299. The IRB letter is uploaded under “other” files, and the statement in the manuscript has been edited for clarification. 

Response: The dataset is owned by the Child and Adolescent Health Measurement Initiative (CAHMI), The Johns Hopkins Bloomberg School of Public Health, Department of Population, Family & Reproductive Health. Data may be requested from https://www.childhealthdata.org/dataset. 

Reviewers' comments:

Reviewer #1: 

This manuscript examines associations between screen-use and obesity, and examines ineractions with physical activity and sleep in a large sample of adolescents from the US. While this is a generally well written manuscript, there are a number of comments for your consideration below:

1. Abstract:

A little untidy with some mistakes/incomplete information. For example, in the introduction you state that you examine effect modification by sleep and PA, then in the methods you state sex and pa. furthermore, you examine the associations between screen time and BMI not the other way around. Consistency in ordering is necessary for the reader to understand what you have done. Nothing in the methods section about how behaviours etc were measured. Results are missing the n and seem quite brief. Consider stating pa not exercise as you measured whether or not adolescents met the pa guidelines, not exercise.

Response: We agree with the reviewers and would like to thank them for pointing out the errors in the abstract. We have updated the abstract to correct the errors and add the missing information. 

2. Introduction:

I found this section lacking in a clear and concise rational.

The data you present on obeisty is quite out of date.

The second paragraph is very disjointed with talk of screen time then sedentary behaviour, but no definition or distinction described.

No rational for sleep or pa as effect modifiers.

I don't think the rational around covid is appropriate here given that your data preceded covid.

Response: We have made following changes to the introduction based on your feedback:

• More recent data on obesity statistics is presented.

• Reorganized the introduction to make it clear

• Removed the description about sedentary behavior

• Included rationale for examining sleep and PA as effect modifiers.

• Removed the mention of COVID-19 from the introduction.

3. Methods:

This section is relatively well described.

We aren't told how covariates were measured and given that you describe them in results, the reader should know how they were measured.

Statistical analysis section is very brief. i would expect more on why you chose this method to look at effect modification and more description. also the ordering seems strange. Typically you would see if your main variables differ by gender at the start and then all subsequent analyses are conducted by gender.

Response: Thank you for your feedback. Based on your suggestions, the following changes were made to the methods section:

• A short description of covariate measurement was added.

• Details were added to the statistical analysis section.

• Interaction terms are commonly used in regression analysis to examine the interaction/effect modification between two risk factors. We presented the results in accordance with the recommendation by Knol and VanderWeele (2012). https://academic.oup.com/ije/article/41/2/514/692957. More details were added to the methods section 

• Table 1 has been updated to include row percentages, which show the distribution of BMI categories by sex. Given the difference in the prevalence of overweight in males and females, we decided to present regression analysis results in the total sample, and stratified by sex. 

4. Results:

This section is too long and the tables could be better presented by having gender as columns not extra rows where variabls are repeated.

First paragraph has a mistake in th % in second sentence.

Response: Thank you for your comment. Phrases and sentences that were repetitive were removed to shorten the section. Table were updated to wide format removing extra rows. 

5. Discussion:

Lacking in discussion and implicatations.

Limitations need expanding to add in that this data is >6 years old and not reflective of current screen-based practices.

Using the phrase exercise is unclear.

Bottom paragraph on page 19 is unclear when you are talking of replacement of pa by screen-time. Your analysis doesnt allow such statements to be made. you could allude to isotemporal substitution, but not suggest that that is what is hapenning here.

No conclusion?

what do your findings mean, to whom and what shall we do now that we have them?

Response: We have edited the discussion section to add the following: 

• Additional discussion of the implications 

• The timing of the survey added as a limitation 

• The sentences related to replacement of PA by screen time were rewritten based on your suggestions

• A conclusion section is added at the end. 

Reviewer #2: 

Bakour and colleagues analyzed the 2016-2017 National Survey of Children’s Health and assessed the association between time spent watching TV, playing video games, using a computer or handheld device, and BMI among U.S. adolescents and potential effect modification of these associations by sleep duration and physical activity. The study has merit but I have several questions/comments:

1. Abstract: Please elaborate what does aOR mean? Then you can use the abbreviation.

Response: Thank you for your comment. We have updated the abstract based on your comment. 

2. Abstract: Line 39: Please write ‘95% CI’ before using 95% confidence interval.

We have updated the abstract based on your comment.

3. After using an odds ratio, please put a semicolon before mentioning 95% CI. For example in Line 39 please use 1.53; 95% CI: 1.19, 1.97.

Response: We have made the recommended change to the abstract.

4. Before using Ordinal logistic regression, was proportional odds assumption verified by the authors?

Response: Thank you very much for your comment. The association between screen time and BMI was examined using multinomial logistic regression model with normal weight as the reference category. The methods section incorrectly listed ordinal instead of multinomial. This error has been correct in the revised manuscript. 

5. In Table 1, please use row percentages instead of column percentages. That will give some idea about the prevalence of each weight categories. 

Response: Thank you for the suggestion. We have updated table 1 to include row percentages, and edited the results section accordingly 

6. In table 2 and 3, please report the unadjusted odds for male and female.

The unadjusted ORs for male and female participants were added to table 2 and 3. 

7. In Page 14, what is the title of the table?

Response: Thank you for your comment. The title was blended with the text of the manuscript. We have updated the text to make sure that the title for the table is clearly visible. 

8. In Table 3, why significant association was found for females but not for males? Please discuss.

Response: Although the association was only significant in females, the estimates were not very different, and the confidence intervals were overlapping, indicating a minor difference in the association between males and females. These findings are consistent with some previous studies, including studies in adults. More details were added to the discussion section.

---

## [Decision Letter · Decision Letter 1]

25 Oct 2022

PONE-D-22-15444R1Association between screen time and obesity in US adolescents: A cross-sectional analysis using National Survey of Children’s Health 2016-2017PLOS ONE

Dear Dr. Bakour,

Thank you for submitting your manuscript to PLOS ONE. After careful consideration, we feel that it has merit but does not fully meet PLOS ONE’s publication criteria as it currently stands. Therefore, we invite you to submit a revised version of the manuscript that addresses the points raised during the review process.

We look forward to receiving your revised manuscript.

Kind regards,

Yih-Kuen Jan, PhD

Academic Editor

PLOS ONE

Journal Requirements:

Reviewers' comments:

Reviewer's Responses to Questions

**Comments to the Author**

1. If the authors have adequately addressed your comments raised in a previous round of review and you feel that this manuscript is now acceptable for publication, you may indicate that here to bypass the “Comments to the Author” section, enter your conflict of interest statement in the “Confidential to Editor” section, and submit your "Accept" recommendation.

Reviewer #1: All comments have been addressed

2. Is the manuscript technically sound, and do the data support the conclusions?

Reviewer #1: Yes

3. Has the statistical analysis been performed appropriately and rigorously? 

Reviewer #1: Yes

4. Have the authors made all data underlying the findings in their manuscript fully available?

Reviewer #1: Yes

5. Is the manuscript presented in an intelligible fashion and written in standard English?

Reviewer #1: Yes

6. Review Comments to the Author

Reviewer #1: The manuscript is much improved following the revisions you have made. I only have a few minor points:

1. in the introduction, you don't tell the reader what screen-time is (line 64) or what the recommendations are that you are referring to (line 67).

2. What are some reasons for these differences (line 99)?

3. Line 133-134, were these categories response options or did you create them? I wondered why you chose to categorise.

7. PLOS authors have the option to publish the peer review history of their article (what does this mean?). If published, this will include your full peer review and any attached files.

Reviewer #1: No

---

## [Author Response · Author response to Decision Letter 1]

6 Nov 2022

Reviewers' comments:

1. in the introduction, you don't tell the reader what screen-time is (line 64) or what the recommendations are that you are referring to (line 67).

Response: We have added a definition of screen time (line 58) and details regarding the recommended screen time per day for adolescents (line 62)

2. What are some reasons for these differences (line 99)?

Response: The studies that found no association between computer use and BMI were conducted in different countries (Germany, Finland, Brazil), while the study that found an association was conducted in the US. A clarification was added to the manuscript (lines 86-88)

3. Line 133-134, were these categories response options or did you create them? I wondered why you chose to categorise.

Response: The two screen time questions in the NSCH had response options of none, less than one hour, one hour, 2-3 hours, 4 hours or more. We combined the first two categories into: None or less than 1 hour, and the middle two categories into 1-3 hours, to preserve sample size, especially when stratifying by sex and physical activity/sleep, and to facilitate presentation of results (including interaction) and interpretation of findings. Clarification was added on line 120.

---

## [Editor Report · Decision Letter 2]

17 Nov 2022

Association between screen time and obesity in US adolescents: A cross-sectional analysis using National Survey of Children’s Health 2016-2017

PONE-D-22-15444R2

Dear Dr. Bakour,

We’re pleased to inform you that your manuscript has been judged scientifically suitable for publication and will be formally accepted for publication once it meets all outstanding technical requirements.

Kind regards,

Yih-Kuen Jan, PhD

Academic Editor

PLOS ONE
---

## [Editor Report · Acceptance letter]

21 Nov 2022

PONE-D-22-15444R2 

Association between screen time and obesity in US adolescents: A cross-sectional analysis using National Survey of Children’s Health 2016-2017 

Dear Dr. Bakour:

I'm pleased to inform you that your manuscript has been deemed suitable for publication in PLOS ONE. Congratulations! Your manuscript is now with our production department. 

Kind regards, 

on behalf of

Dr. Yih-Kuen Jan 

Academic Editor

PLOS ONE